# Disentangling the Roles of Distinct Cell Classes with Cell-Type Dynamical Systems

**Aditi Jha**
Electrical and Computer Engineering
Princeton University
aditijha@princeton.edu

**Diksha Gupta**
Sainsbury Wellcome Center
University College London
diksha.gupta@ucl.ac.uk

**Carlos D. Brody**
Princeton Neuroscience Institute
Princeton University
brody@princeton.edu

**Jonathan W. Pillow**
Princeton Neuroscience Institute
Princeton University
pillow@princeton.edu

## Abstract

Latent dynamical systems have been widely used to characterize the dynamics of neural population activity in the brain. However, these models typically ignore the fact that the brain contains multiple cell types. This limits their ability to capture the functional roles of distinct cell classes, and to predict the effects of cell-specific perturbations on neural activity or behavior. To overcome these limitations, we introduce the "cell-type dynamical systems" (CTDS) model. This model extends latent linear dynamical systems to contain distinct latent variables for each cell class, with biologically inspired constraints on both dynamics and emissions. To illustrate our approach, we consider neural recordings with distinct excitatory (E) and inhibitory (I) populations. The CTDS model defines separate latents for both cell types, and constrains the dynamics so that E (I) latents have a strictly positive (negative) effects on other latents. We applied CTDS to recordings from rat frontal orienting fields (FOF) and anterior dorsal striatum (ADS) during an auditory decision-making task. The model achieved higher accuracy than a standard linear dynamical system (LDS), and revealed that the animal's choice can be decoded from both E and I latents and thus is not restricted to a single cell-class. We also performed in-silico optogenetic perturbation experiments in the FOF and ADS, and found that CTDS was able to replicate the experimentally observed effects of different perturbations on behavior, whereas a standard LDS model—which does not differentiate between cell types—did not. Crucially, our model allowed us to understand the effects of these perturbations by revealing the dynamics of different cell-specific latents. Finally, CTDS can also be used to identify cell types for neurons whose class labels are unknown in electrophysiological recordings. These results illustrate the power of the CTDS model to provide more accurate and more biologically interpretable descriptions of neural population dynamics and their relationship to behavior.

## 1 Introduction

Advancements in neural recording technologies have made it possible to record from hundreds of neurons simultaneously [32–34]. Understanding the dynamics of these high-dimensional populations and their relationship to complex behavior is a fundamental goal of computational neuroscience. Dynamical systems have proven to be extremely useful in this pursuit [20, 3, 11, 12]. Latent dynamical

38th Conference on Neural Information Processing Systems (NeurIPS 2024).

systems model neural activity as arising from a low-dimensional latent state, and describe how the activity evolves as a function of time in this low-dimensional space.

However, while latent dynamical systems provide parsimonious descriptions of neural activity, they fail to capture key functional and biological properties of real neural circuits. In particular, they ignore the fact that neural circuits consist of multiple cell types. Standard models describe the activity of all neurons as arising from the same set of latent variables, without any differences among cell types or constraints on the interactions between them. This prevents latent dynamical systems from shedding light on the functional roles of distinct cell classes [27, 36, 28, 26]. Furthermore, it prevents them from accurately describing optogenetic perturbation experiments, in which specific cell classes within a neural circuit are perturbed. This severely limits the usefulness of classic latent dynamical systems models for identifying the causal effects of neural activity on behavior [15, 24].

To overcome these limitations, we introduce the "cell-type dynamical systems" (CTDS) model. This model extends latent linear dynamical system (LDS) models by assigning a unique set of latent variables to control the activity of each cell class, while also imposing constraints on the model parameters that dictate the sign of interactions between different cell types. Here we focus on networks with two distinct cell types: excitatory (E) and inhibitory (I) neurons. Standard analyses would characterize the dominant modes of network activity without considering the distinct roles played by E and I cells. By contrast, the CTDS model defines separate latents for E and I cells, and constrains the dynamics matrix so that E and I latents have a positive or negative (respectively) effect on the other latents. Additionally, we also constrain the mapping from the latent space to the observation space to be non-negative, thus projecting Dale's law from the dynamics to the observation space. Our model also easily extends to multi-region settings, and allows us to explicitly constrain the types of cells (e.g., E cells) making long range projections.

We first demonstrate an equivalence between CTDS models with E and I cells and low-rank recurrent neural networks composed of E and I neurons (EI-RNN), both theoretically and using simulations. EI-RNNs [13, 31, 29] are widely studied as proxies of neural circuits, thus this equivalence validates the utility of our model for understanding neural circuits. Next, we apply CTDS to recordings from rat frontal orienting fields (FOF) and anterior dorsal striatum (ADS) during an auditory decision-making task [4]. We show that CTDS predicts neural activity better on held out trials than a standard LDS model; a multi-region CTDS with distinct latents for FOF and ADS furthermore outperforms CTDS with no regional constraints. We show that the latents extracted from both E and I populations encode the animal's choice, revealing that choice information is not restricted to a single cell class, consistent with recent findings [23]. Additionally, we show that a classifier trained on the inferred latent states during training trials predicts the animal's behavioral choice well during test trials.

Next, we use CTDS to perform in-silico optogenetic perturbation experiments modeled after those performed in vivo. In these experiments, one class of neurons (E or I) in a single brain region is perturbed during a portion of each trial. Remarkably, we find that the CTDS model—despite not being trained on perturbation data—can accurately predict the effects of different perturbations on behavior. Moreover, CTDS allows us to visualize the effects of these perturbations on the underlying dynamics of different regions, providing precise insights into the roles of distinct regions and cell-types. A standard LDS, however, is unable to replicate experimentally observed findings during the same set of perturbations due to the lack of cell information and appropriate structural constraints. Finally, we show that CTDS can be used to identify the cell class of neurons whose type is unknown in experimental recordings. While in previous analyses we assumed that the identity of neurons were known, cell-type information is often not known in neurophysiology experiments. We developed an approach to infer the identities of unknown cells using CTDS, and find that CTDS is indeed able to infer cell-types of upto $50\%$ of unidentified neurons using recordings from the FOF and ADS. This is an exciting application of our model as tagging cell-types can be challenging during the course of real-world experiments. Overall, our findings underscore that CTDS provides a versatile tool to study the effects of perturbations on neural dynamics, as well to infer cell identities.

## 2 Cell-type latent linear dynamical systems (CTDS)

### 2.1 Background: LDS models

A latent linear dynamical system (LDS) model describes the firing activity of a population of $N$ neurons at time $t$, $\mathbf{y}_t \in \mathbb{R}^N$, as arising from a low-dimensional latent state $\mathbf{x}_t \in \mathbb{R}^D$. This state

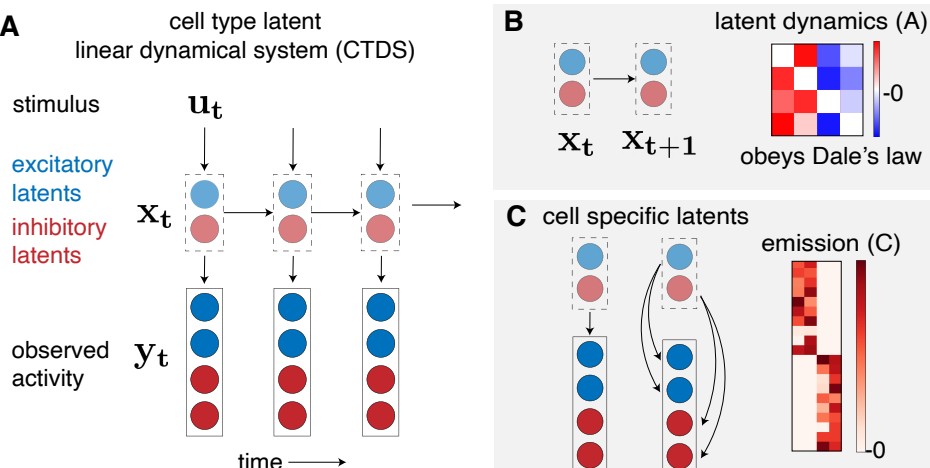

Figure 1: **A.** Graphical model of a Cell-Type latent Dynamical System (CTDS). Red and blue circles represent the observed activity, $\mathbf{y}_t$, of excitatory and inhibitory neurons respectively at time $t$. The color of the latents $\mathbf{x}_t$ reflects the cell-type they govern. $\mathbf{u}_t$ is any input to the system at the time $t$. **B.** Latent dynamics, $A$, obeys Dale's law, such that the columns of $A$ are either non-negative or non-positive corresponding to E and I latents. **C.** Distinct sets of latents govern the activity of E and I neurons. The emission matrix, $C$, is also constrained to be non-negative such that the Dale's law constraint in the latent space is reflected to the neural activity space.

evolves according to linear dynamics and under the influence of any external input $\mathbf{u}_t \in \mathbb{R}^M$:

$$\mathbf{x}_{t+1} = A\mathbf{x}_t + B\mathbf{u}_t + \mathbf{w}_t; \ \mathbf{w}_t \in \mathcal{N}(\mathbf{0}, Q) \tag{1}$$

Here, $A \in \mathbb{R}^{D \times D}$ captures dynamics in the latent space, $B \in \mathbb{R}^{D \times M}$ captures the influence of the external input, while $Q \in \mathbb{R}^{D \times D}$ is the noise covariance. The observed activity is then a linear transformation of this latent state such that

$$\mathbf{y}_t = C\mathbf{x}_t + \mathbf{v}_t, \ \mathbf{v}_t \in \mathcal{N}(\mathbf{0}, R) \tag{2}$$

Here, $C \in \mathbb{R}^{N \times D}$ is called the emission matrix, and $R \in \mathbb{R}^{N \times N}$ is the observation noise covariance.

## 2.2 The CTDS model

To disentangle the roles of distinct cell types in a population, we modify the standard LDS model in two ways: (1) we define a distinct set of latent variables for the activity of each cell class; and (2) we constrain the dynamics to obey functional properties of these cell types. We refer to the resulting models as Cell-Type latent linear Dynamical Systems (CTDS) [1]. Here we focus on the setting of excitatory (E) and inhibitory (I) cell types, as schematized in Fig. 1. We use distinct sets of latents for E and I neurons, $\mathbf{x}_t^e \in \mathbb{R}^{D_e}$ and $\mathbf{x}_t^i \in \mathbb{R}^{D_i}$ respectively, such that $D_e + D_i = D$ (shown in Fig. 1C). As a result, the emission matrix $C$ is block-diagonal with blocks of shape $N_e \times D_e$ and $N_i \times D_i$—the first block maps the E latents to the $N_e$ excitatory neurons, and the second block maps the I latents to the $N_i$ inhibitory neurons. Note that we constrain these blocks to be non-negative, thus ensuring that an increase or decrease in a given latent maps to an increased or decreased firing rate (respectively) in neurons of the associated cell type.

Next, to ensure consistency with Dale's law—E neurons excite and I neurons inhibit—we constrain the sign of the columns of the dynamics matrix $A$ so that columns associated with E latents are non-negative and columns associated with I latents are non-positive (Fig. 1B). (Note we do not apply this constraint to the diagonal elements of $A$; this allows individual latents to have positive auto-correlation and evolve smoothly in time, regardless of cell type).

---

[1]Code available here.

## 2.3 Multi-region CTDS

Given simultaneously recorded neural activity from multiple brain regions, we often want to understand the computations performed by each region as well as the interactions between regions. To do so, following existing work on multi-region modeling [12], we extend our model to multiple regions such that each region has its own set of latents. For example, if we have $K$ regions such that $\mathbf{x}_t^k \in \mathbb{R}^{D_k}$ is the latent vector for region $k$, this latent state evolves as follows:

$$\mathbf{x}_t^k = A_{kk}\mathbf{x}_{t-1}^k + \sum_{j \neq k} A_{jk}\mathbf{x}_{t-1}^j + B\mathbf{u}_t + \epsilon_t, \tag{3}$$

where $A_{kk} \in \mathbb{R}^{D_k \times D_k}$ captures the within-region dynamics of the $k$th region, while $A_{jk} \in \mathbb{R}^{D_j \times D_k}$ models the inter-region communication from region $j$ to region $k$. Since we are interested in cell-type specific information, we further assume separate latents for distinct cell types within a region such that $\mathbf{x}_t^k = [\mathbf{x}_t^{k,e}, \mathbf{x}_t^{k,i}]$. The within-region dynamics $A_{kk}$ obeys Dale's law, similar to the single-region model. Furthermore, since long range connections in the cortex are generally known to be excitatory [30], we can also constrain the inter-region communication matrix $A_{jk}$ to have non-negative columns corresponding to the E latents of region $j$, and zero for the I latents. Finally, to ensure that cell-type specific and region-specific latents only control their respective neurons, the emission matrix $C \in \mathbb{R}^{N \times KD}$ is non-negative and block diagonal. Mathematically, we can write the activity of E and I cell classes in region $k$ as follows:

$$\mathbf{y}_t^{k,e} = C^{k,e}\mathbf{x}_t^{k,e} + \mathbf{v}_t^{k,e}, \ \mathbf{v}_t^{k,e} \in \mathcal{N}(\mathbf{0}, R^{k,e}) \tag{4}$$

$$\mathbf{y}_t^{k,i} = C^{k,i}\mathbf{x}_t^{k,i} + \mathbf{v}_t^{k,i}, \ \mathbf{v}_t^{k,i} \in \mathcal{N}(\mathbf{0}, R^{k,i}). \tag{5}$$

## 2.4 Inference procedure

In order to infer the parameters of a CTDS model, we maximize the expected log-likelihood of observed data under the model using the Expectation Maximization (EM) algorithm [6]. We will describe inference for the single-region model for simplicity, which can easily be extended to the multi-region variant. We want to infer the model parameters, $\Theta = \{A, B, C, Q, R\}$, given $N$ trials of observed data $\{\mathbf{y}_{1:T_n}^n\}_{n=1}^N$ where $n$ represents the trial index.

In the Expectation step, we perform standard Kalman filtering and smoothing, to learn posterior distributions of the latent states $P(\mathbf{x}_t^n \mid \mathbf{y}^n, \Theta)$, and $P(\mathbf{x}_t^n, \mathbf{x}_{t+1}^n \mid \mathbf{y}_{1:T_n}^n, \Theta) \ \forall t \in \{1, T_n\}$. In the Maximization step, we learn the model parameters given the computed expectations of the latent states. Specifically, we optimize the following expression

$$\mathcal{L}_{CD}(\Theta) = -\frac{1}{2}\sum_{n=1}^N \left( \mathbb{E}\Big[ \sum_{t=1}^{T_n-1} \log \mathcal{N}(\mathbf{x}_{t+1}^n; A\mathbf{x}_t^n + B\mathbf{u}_t^n, Q) + \sum_{t=1}^{T_n} \log \mathcal{N}(\mathbf{y}_t^n; C\mathbf{x}_t^n, R)\Big] \right) \tag{6}$$

for $\{A, B, C, Q, R\}$, such that $A$ and $C$ obey their respective structural constraints. As a result, unlike a standard LDS, we no more have closed form updates for the model parameters. We instead solve two quadratic programs under the constraints on $A$ and $C$ separately:

$$\max_{A,B} -\frac{1}{2}\sum_{n=1}^N \left( \mathbb{E}\Big[ \sum_{t=1}^{T_n-1} \log \mathcal{N}(\mathbf{x}_{t+1}^n; A\mathbf{x}_t^n + B\mathbf{u}_t^n, Q)\Big] \right) \text{ s.t. } A \text{ obeys Dale's law} \tag{7}$$

$$\max_{C} -\frac{1}{2}\sum_{n=1}^N \left( \mathbb{E}\Big[ \sum_{t=1}^{T_n} \log \mathcal{N}(\mathbf{y}_t^n; C\mathbf{x}_t^n, R)\Big] \right) \text{ s.t. } C >= 0 \text{ and block-diagonal} \tag{8}$$

We alternate between optimizing $A, B, C$ and the noise matrices $\{Q, R\}$, which have closed form expressions once $\{A, B, C\}$ are fixed. Overall, we alternate between the expectation and maximization steps until the log-likelihood converges. We use the MOSEK solver [2] with CVXPY [7, 1] to perform the quadratic optimizations for the constrained matrices $\{A, C\}$.

## 3 Relationship to E-I recurrent neural networks

Recurrent neural networks (RNNs) are widely used to study neural dynamics in the brain [21, 25, 8, 13, 31]. An important line of work has focused on RNNs with excitatory and inhibitory units,

which mirror the functional organization of real neural circuits [10, 13, 24, 31, 29]. However, such models are often hard to analyze or interpret. Here we show that CTDS provides a natural bridge between EI-RNNs and latent dynamical systems, in which neural activity is explained in terms of a low-dimensional latent variable. Our work thus also provides an explicit connection to low-rank RNNs, which have been shown to account for a wide variety of tasks and datasets [22, 8, 35], but which have not to our knowledge been equipped with distinct cell types. Specifically, we show that under certain conditions a low-rank EI-RNN is formally equivalent to a CTDS, and we derive the mapping from one model class to the other.

Let $\mathbf{y}_t \in \mathbb{R}^N$ be the activity generated by a linear RNN containing $N$ neurons, such that:

$$\mathbf{y}_{t+1} = J\mathbf{y}_t + \eta_t, \quad \eta_t \sim \mathcal{N}(0, P), \tag{9}$$

where $J \in \mathbb{R}^{N \times N}$ obeys Dale's law, with non-negative columns for E cells and non-positive columns for I cells, and $P \in \mathbb{R}^{N \times N}$ is the noise covariance. Let's further assume $N_e$ number of E cells, and $N_i$ number of I cells. For simplicity, we will focus here on the noiseless case, $P = \mathbf{0}$; we discuss the noisy case in the supplement (sec. A1).

We can further write the activity of the E and I cells separately as follows:

$$\mathbf{y}_{t+1}^e = J_{ee}\mathbf{y}_t^e + J_{ei}\mathbf{y}_t^i, \quad \mathbf{y}_{t+1}^i = J_{ii}\mathbf{y}_t^i + J_{ie}\mathbf{y}_t^e \tag{10}$$

where $J_{ee}$ and $J_{ie}$ represent blocks of $J$ that contain outgoing weights from E cells and are thus non-negative, $J_{ii}$ and $J_{ei}$ are blocks of outgoing weights from I cells and contain non-positive elements. Let's define a new matrix $J^+$ which contains the absolute values of $J$ such that:

$$J^+ = \begin{bmatrix} J_{ee} & J_{ei}^+ \\ J_{ie} & J_{ii}^+ \end{bmatrix} \tag{11}$$

Here, $J_{ii}^+$ and $J_{ie}^+$ contain absolute values of the all-negative matrices $J_{ii}$ and $J_{ie}$, respectively. Mapping this RNN to a CTDS with E and I cells requires that the top and bottom sub-matrices of $J^+$ have low-rank non-negative matrix factorization (NNMF) solutions:

$$\begin{bmatrix} J_{ee} & J_{ei}^+ \end{bmatrix} = U_1 V_1^\top; \quad \begin{bmatrix} J_{ie} & J_{ii}^+ \end{bmatrix} = U_2 V_2^\top \tag{12}$$

where $U_1 \in \mathbb{R}^{N_e \times K_1}$, $V_1 \in \mathbb{R}^{N \times K_1}$ and $U_2 \in \mathbb{R}^{N_i \times K_2}$, $V_2 \in \mathbb{R}^{N \times K_2}$. Thus, $J^+$ can be written as:

$$J^+ = UV^\top = \begin{bmatrix} U_1 & \mathbf{0} \\ \mathbf{0} & U_2 \end{bmatrix} \begin{bmatrix} V_1^\top \\ V_2^\top \end{bmatrix} \tag{13}$$

where $U, V \in \mathbb{R}^{N \times (K_1 + K_2)}$ are non-negative matrices. If we make the rows of $V$ corresponding to I cells negative, and call this new matrix $V_{\text{dale}}$, we obtain $J = UV_{\text{dale}}^\top$. This EI-RNN can then be equivalently be written as a CTDS of rank at most $K_1 + K_2$ as follows [35]:

$$\mathbf{x}_t = V_{\text{dale}}^\top U \mathbf{x}_{t-1} \tag{14}$$
$$\mathbf{y}_t = U\mathbf{x}_t, \tag{15}$$

with the emissions $C = U$, and latent dynamics $A = V_{\text{dale}}^\top U$. Both $C$ and $A$ obey the required constraints: $C$ is non-negative and block diagonal, and $A$ obeys Dale's law. Hence, in summary, a linear EI-RNN can be mapped to a CTDS if the excitatory and inhibitory rows of the absolute connectivity matrix, $J^+$, have low-rank NNMF solutions. In the case of non-zero noise in the EI-RNN, it is still possible to find a mapping to a cell-type LDS model, however that requires further constraints on the connectivity and noise structure of the RNN (see sec. A1).

We also generated activity from E-I networks whose connectivity $J$ obeys the above constraints, and show that CTDS recovers the connectivity accurately—with smaller error than a standard LDS using the same amount of data (see sec. A2). Finally, when fitting CTDS models to real data, we use insights from this equivalence to initialize them. In particular, we first learn a $J$ matrix that obeys Dale's law by solving a contrained regression problem given the data $\{\mathbf{y}_{1:T}\}$:

$$\mathbf{y}_{t+1} = J\mathbf{y}_t + \mathbf{v}_t; \quad \text{s.t } J \text{ obeys Dale's law} \tag{16}$$

We apply NNMF as discussed earlier on the absolute values of the learned $J$ per eq. 13, and thus initialize $C = U$, $A = V_{\text{dale}}^\top U$.

# 4 Application to rodent decision-making data

Next, we applied the CTDS model to neural data collected from rats trained to perform an auditory decision-making task [4]. On each trial, animals heard randomly timed clicks from the left and the right, and were rewarded for selecting the side with more clicks (Fig. 2A). Neuropixel probes were used to record simultaneous neural activity from two regions known to be involved in evidence accumulation: the frontal orienting fields (FOF) and the anterior dorsal striatum (ADS) [14, 37]. Previous work has shown that individual neurons in FOF and ADS exhibit side selectivity in this task, with activity ramping upwards for stimuli on the cell's preferred side [14, 37]. However, to carefully understand the causal role of each region during evidence accumulation (e.g., whether regions carry out different aspects of the task), it is important to disentangle the roles of different cell types in each region. We point out that prior work has analyzed the two regions independently, while here we also study communication between them.

## 4.1 Model fitting details

We fitted a standard LDS and the CTDS model to zero-centered firing rates of 109 active neurons recorded in FOF and ADS regions of a single rat. These neurons were filtered to have a minimum firing rate of 1Hz during the ($\sim$1s) stimulus period. We used 50ms bins to obtain firing rates from the spiking activity of neurons, and zero-centered the response of each neuron across trials. We had access to 353 trials for this animal. To fit CTDS models, we labeled neurons as E or I using both anatomical information about the two regions as well a clustering of spike width histograms (see sec. A3). Additionally, since we had data from two regions, we used region identity and also fit multi-region CTDS models. Thus, we also added region-specific constraints on the dynamics matrix of this model—within-region dynamics obeyed Dale's law. Since ADS is a sub-cortical region and the pathway from ADS to FOF is multi-synaptic, we did not put any constraints on cross-region communication from ADS to FOF. However, we restricted communication from FOF to ADS to be excitatory (as discussed in subsec. 2.3).

We used 80% of 353 trials to fit our models, the remaining 20% trials were held out. The models also received a 2-dimensional input at every time point, containing the number of left and right clicks played between the previous and current time points. We initialized CTDS models using the NNMF initialization process described in sec. 3. For all models, we used 10 initialization seeds and picked the best seed based on training log-likelihood. Finally, we varied the number of dimensions in the latent space of the models, and for simplicity used the same number of latents for each cell type (note that ADS has inhibitory neurons only, so we only used I latents for this region when fitting CTDS models).

## 4.2 Both cell types encode choice information

We found that CTDS and its multi-region variant outperformed LDS models in log-likelihood of held-out trials. Fig. 2B shows test log-likelihood as a function of the number of latents available to each cell-type. This result confirms that adding E-I structure to latent linear dynamical systems allows the model to capture neural data well, while also providing additional interpretability. For each model class, we also trained a logistic regression classifier to predict the animal's choice from the last time-step of the latent state. Fig. 2C shows the choice prediction accuracy of the classifier corresponding to each model on test trials (for $D = 6$ per population). We observe that all models are able to predict choice well (and far above chance, i.e. 50%).

Next, we analyzed a fitted multi-region CTDS model to draw scientific insights about the roles of the two regions and cell types. We chose the model with 6 dimensions per cell-type for interpretability, resulting in overall 18 latents across the two regions and cell-types (E and I for FOF, I for ADS). In Fig. 2D, we show the recovered dynamics matrix—our results reveal cross-region communications between FOF and ADS in both directions and suggesting that the regions are recurrently connected during evidence accumulation tasks.

Finally, we plotted the top two PCs of the inferred latent states for FOF and ADS in Fig. 2E, with left and right choice trials colored distinctly. We found that both E and I latents in the FOF encoded the animal's choice, with their trajectories separating out for left and right choices in opposite directions. This is consistent with recent work from Najafi et al. [23], which showed that both excitatory and

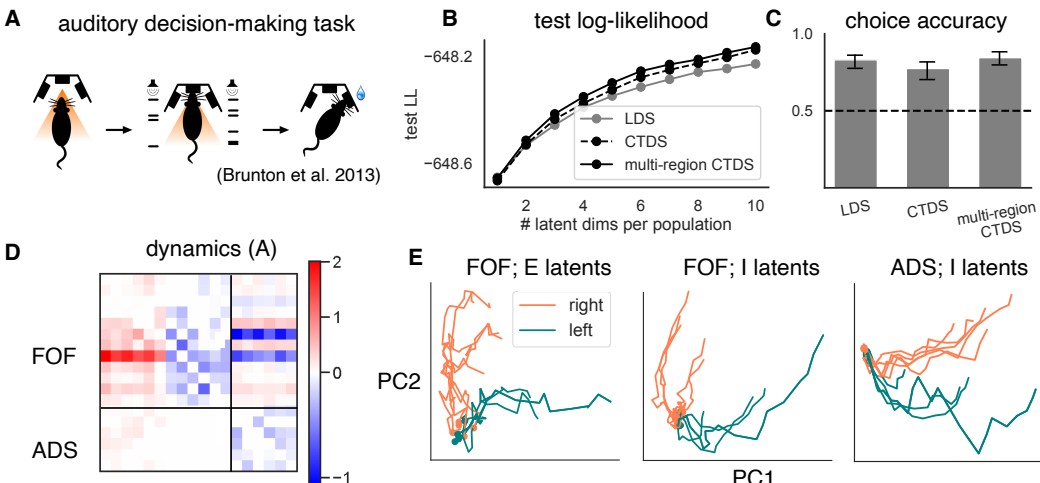

Figure 2: CTDS applied to decision-making task in rodents. **A**. "Poisson clicks" auditory decision-making task [4] **B**. Log-likelihood on held out trials as a function of the number of latents per cell-type, for three different model classes. LDS (gray), CTDS with no region structure (dashed black), multi-region CTDS (solid black). **C**. Choice accuracy (between 0–1) on test trials using classifiers trained on latents inferred from the three models (with six latents per cell-type). Dotted line represents chance performance. Error bars show one standard deviation across 10 different sampled latent states. **D**. Recovered dynamics matrix, $A$, using a multi-region CTDS with 6 latents per cell-type. The within-region dynamics matrix in both regions (top left: FOF, bottom right: ADS, note that ADS has inhibitory neurons) obey Dale's law, with columns being either excitatory only or inhibitory only (we zeroed out the diagonal for visualization). **E**. Latent state trajectories plotted along their top two PCs colored distinctly for left (coral) and right (teal) choice trials. (Left) E and I latents in FOF, and (right) I latents in ADS. The trajectories are well-separated for left and right trials.

inhibitory neurons are equally selective to choice. We also find that ADS latents encode choice information. Thus, CTDS enabled us to disentangle and understand the dynamics underlying distinct classes of cells and regions.

## 4.3    In-silico optogenetic perturbation experiments

By dissociating dynamics underlying each cell-type, CTDS models allow us to study cell-specific optogenetic perturbations. These experiments involve activating or silencing a targeted class of neurons in the brain to causally establish links between neural circuits and observed behavior. We can perform in-silico perturbation experiments in CTDS models by perturbing latents corresponding to a particular cell class, and consequently study the effects of these perturbations on both neural population dynamics and behavior of the animal.

During the clicks task, previous work by Hanks et al. [14] has shown that silencing excitatory neurons in FOF during the the first half of the trial does not affect the animal's behavior. However inactivation during the second half produces biased choices and reduced task accuracy. Furthermore, Yartsev et al. [37] have shown that inactivating inhibitory ADS neurons results in behavior deficits during both early and late halves of the trial.

To investigate the circuit-level origins of these effects, we conducted in-silico perturbations using our multi-region CTDS model from Fig. 2 (with 6 latent variables per cell type), which was modeled after in-vivo perturbations. Importantly, the model was trained only on unperturbed data and did not see perturbed trials during training.

We provided the model with inputs representing left and right clicks per time bin, generated using Poisson processes. Next, we simulated perturbed data by clamping the cell-specific latents in either region to a negative value during the first or second half of the trial. This effectively suppressed the activity of the corresponding cell class (the strength of the inactivation was determined through a parameter search, see Sec. A4). In total, we generated 200 perturbed trials with equal left and right

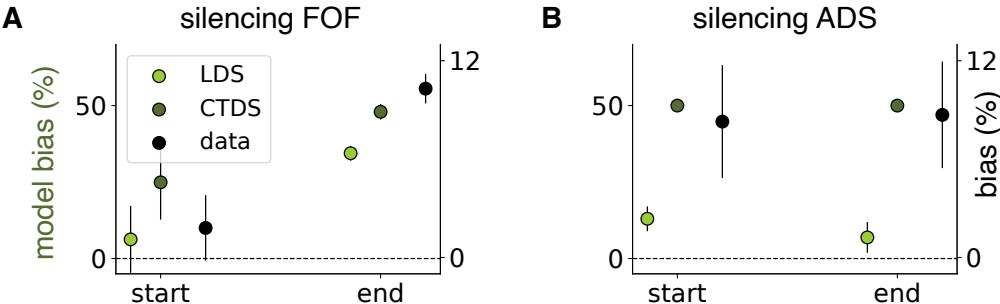

Figure 3: In-silico perturbations using fitted multi-region CTDS and LDS models. **A.** The right y-axis displays the experimentally observed ipsilateral bias from in-vivo perturbations performed by [14] The left y-axis shows the behavioral bias predicted by the CTDS and LDS models when silencing FOF E neurons during the early and late halves of a trial, relative to the correct choice for that trial. For the LDS model, 50% of FOF latents were randomly silenced, as this model does not distinguish cell types. **B**. Similar to **A**, but with ADS I neurons silenced during the early and late halves of trials. Separate y-axes are used because the magnitude of model inactivation was not tuned to match the experimental bias, though the relative bias changes are well captured by the CTDS model. Error bars on the left y-axes represent one standard error across 10 different sampled latents from the fitted models, while those on the right y-axis show one standard error across all animals.

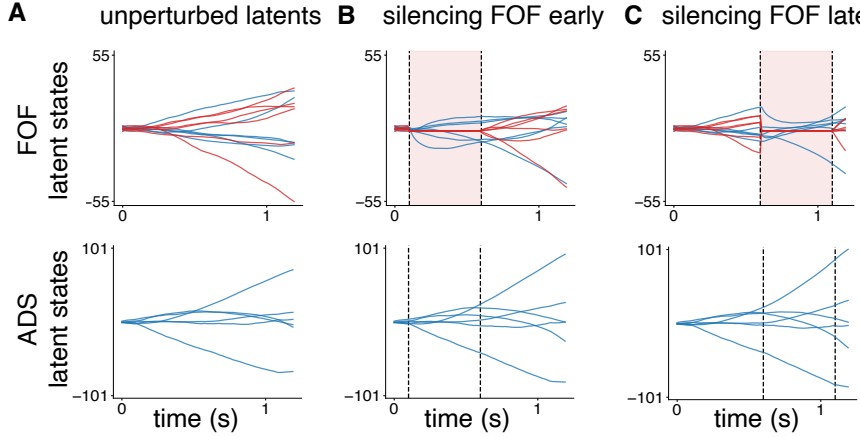

Figure 4: Latent states of fitted CTDS during FOF perturbations. **A.** *(Top)* Trial-averaged unperturbed FOF latents, with E latents in red and I latents in blue. *(Bottom)* Trial-averaged unperturbed ADS latents. **B.** *(Top)* FOF latents and *(bottom)* ADS latents, both trial-averaged, when FOF E latents are silenced during the early half of trials. **C.** *(Top)* FOF latents and *(bottom)* ADS latents, both trial-averaged, with FOF E latents silenced during the later half of trials.

click conditions. We then predicted the choices on these perturbed trials using the classifier that had been previously trained on the unperturbed data.

Remarkably, we found that the perturbed CTDS model produced the same pattern of deficits in FOF as observed in previous experimental work [14](Fig. 3A). Early-half perturbations produced low bias in both the CTDS model and in real animal experiments. On the contrary, late-half perturbations indeed resulted in biased animal behavior, with high ipsilateral bias reported from both the model and the experimental results [14]. We also performed in-silico perturbations in the ADS by inactivating I latents during either early half or later half of the trial. Excitingly, we were again able to replicate experimental findings ( Fig. 3B). Behavioral bias was high when inactivating ADS during both early half and late half, suggesting that animal behavior changes due to perturbations during both halves of a trial. These results demonstrate that the CTDS model successfully replicates the findings from optogenetic inactivation experiments.

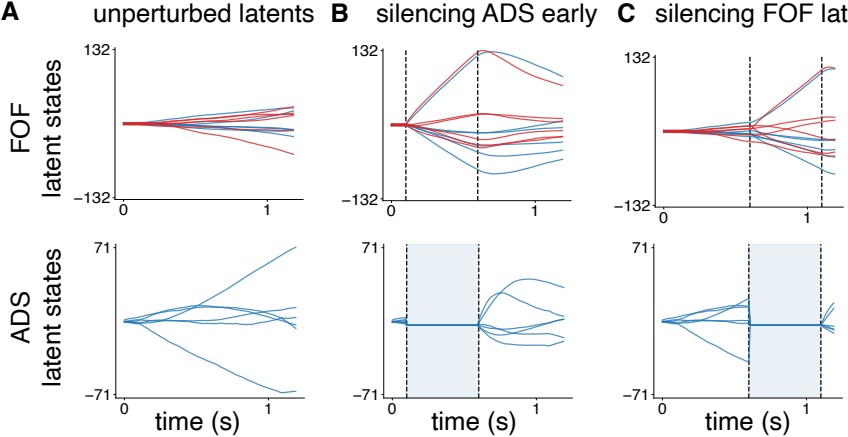

Figure 5: Latent states of fitted CTDS during ADS perturbations. **A.** *(Top)* Trial-averaged unperturbed FOF latents, with E latents in red and I latents in blue. *(Bottom)* Trial-averaged unperturbed ADS latents. **B.** *(Top)* FOF latents and *(bottom)* ADS latents, both trial-averaged, when ADS latents are silenced during the early half of trials. **C.** *(Top)* FOF latents and *(bottom)* ADS latents, both trial-averaged, with ADS silenced during the later half of trials.

This motivated us to further use CTDS to visualize changes in the underlying dynamics and to understand the differential effects of inactivations in FOF and ADS. We visualized the unperturbed latent states averaged across trials, and compared them with the perturbed latent states during the early and late phases of the same trials. We see in Fig. 4B (top) that FOF latents recovered quickly after inactivations during early half of the trial (presumably using information from ADS, Fig. 4B (bottom)), and as a result we did not observe behavior deficits in this case. However, silencing FOF during later half of the trial caused the latent states of FOF to shrink, resulting in the observed behavioral deficits. Next, when perturbing ADS (Fig. 5B&C), the latent dynamics changed significantly during both early and late stage inactivations explaining the behavioral deficits observed during inactivation in ADS. Thus, these results suggest that ADS is crucially involved during the task throughout, and perturbing this region results in loss of choice information in both regions, ultimately resulting in biased animal behavior.

For comparison, we also trained a multi-region LDS model on the same dataset, again with $D = 6$ latents per region. This model had no explicit representation of cell-types, but it did contain distinct latents for each brain region. We attempted to replicate the perturbation experiments in this model by clamping three of the six latents in each region to a negative value (just as we did with CTDS, although in this case the latents had no explicit assignment to distinct cell types). As before, we then used the perturbed model activity to predict choice using a classifier trained on unperturbed trials. This model accurately captured behavior on unperturbed trials (Fig. 2C shows error between model predictions and animal behavior). However, it notably failed to capture the pattern of deficits observed in perturbed trials (Fig. 3A&B). In particular, perturbations in ADS during both halves of a trial resulted in a small behavioral bias, similar to early half FOF perturbations (also see subsec. A5 for latent state visualizations). This is distinct from experimental results, which observed biased behavior during ADS perturbations but not during early-half FOF perturbations. This shows that standard LDS did not accurately capture the functional contributions of different cell-types to neural population dynamics. These results underscore the importance of cell-type models for understanding perturbation experiments, and for uncovering the contributions of different cell types to population dynamics.

## 5   Inferring cell-type information of unknown neurons

Classifying neurons into cell-types facilitates scientific discovery, provides a nuanced view of neural circuits engaged in tasks, and is also crucial for analyses of diseases [38]. However, identifying cell-types is challenging. For example, spike width is commonly used to classify putative excitatory or inhibitory neurons, but many neurons exhibit intermediate spike widths, and thus cannot be

**A** cell-type identification

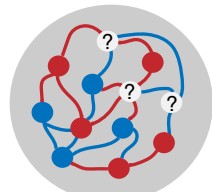

**B** accuracy of cell-types

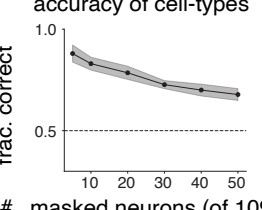

\# masked neurons (of 109)

Figure 6: Cell-type identification. (**A**) Schematic of unknown cell type problem. (**B**) Accuracy of cell-type inference as a function of the number of masked neurons. Error bars show one standard error over 10 initialization seeds. Dotted line represents random performance.

confidently assigned to one class or the other. Previous work has used linear dynamical systems to cluster neurons based on cell-identity [5, 16]. Here we describe a method for using the CTDS model to infer cell-types from neural activity for a subset of neurons in a recording.

Our proposed method involves first fitting the CTDS model using only neurons with known cell type. Then, we run an altered version of the M-step, in which we select for each neuron which set of latents best describe its activity (e.g., E latents or I latents). Specifically, for an unknown $k$-th neuron, assuming diagonal observation noise $R$, we solve the following two regression problems:

$$\min_{C^k} \sum_{n=1}^{N} \mathbb{E}\big[\mathbf{y}_t^{n,k} - C^k \mathbf{x}_t^e - \mathbf{v}_t^k\big], \quad \min_{C^k} \sum_{n=1}^{N} \mathbb{E}\big[\mathbf{y}_t^{n,k} - C^k \mathbf{x}_t^i - \mathbf{v}_t^k\big] \quad \text{s.t} \ C^k >= 0 \qquad (17)$$

Here $C^k \in \mathbb{R}^D$, where $D$ is the latent dimensionality of each cell-type. This step is repeated within each $M$ step during fitting for each unknown neuron. We then pick the $C^k$ that resulted in minimum regression error in the end, thus obtaining the identity of the neuron.

To test our method, we masked between $5 - 50$ neurons (equal number of E and I cells) from the FOF-ADS data, and inferred their identities while fitting a multi-region CTDS ($D = 6$ latents per cell-type). Fig. 6 shows that we were indeed able to infer cell identities well above chance, thus demonstrating the usefulness of CTDS for inferring of cell identities.

## 6   Conclusions and discussion

We have developed a novel framework for disentangling the roles of distinct cell-types in neural circuits. In particular, we extended linear dynamical systems to incorporate cell-type specific information. We focused on excitatory and inhibitory cell classes, with latents constrained to interact in accordance with Dale's law.

We have also derived a theoretical equivalence between linear RNNs with an E-I structure and our model. Cell-type dynamical systems thus bridge a gap between interpretable state-space models and mechanistic models of recurrent computation. We extended CTDS to incorporate multiple regions, resulting in multi-region CTDS models. Application of our model to decision-making data from rodents revealed that including E-I structure in latent linear dynamical systems improved their ability to capture neural activity, and allowed us to understand roles of distinct cell classes, In particular, in line with Najafi et al. [23], we found that both E and I neurons encode choice information.

Crucially, CTDS allowed us to replicate optogenetic perturbation experiments in the FOF and ADS [14, 37], due to the separation of latents based on cell types. Our model predicted the same changes in behavior as observed in experimental studies, and also allowed us to visualize and understand the effects of these perturbation on the dynamics of different cell-typess. This is particularly exciting as a standard LDS was unable to capture the same causal effects. We also developed an approach using CTDS to infer the cell-type information of neurons using their activity.

Our current model assumes linear dynamics, however the core idea of separating latents into cell types is applicable to non-linear systems as well [17, 18] and should be explored in future work. Future studies should explore this extension to better capture the complexity of neural systems. Additionally, our model focuses primarily on populations containing two broad cell classes: excitatory and inhibitory neurons. However, neural populations can be subdivided into more fine-grained cell types, and we aim for our model to be applicable to studying these more specific cell types in the future.

Overall, we believe that CTDS can be broadly useful for several applications, and can serve as a helpful tool to obtain a nuanced view of the dynamics underlying different populations of neurons in the brain.

# 7 Acknowledgements

We thank Anushri Arora, Benjamin Cowley, Lenca Cuturela and Yoel Sanchez Araujo for useful discussions and feedback at various points in the project. We thank the anonymous NeurIPS reviewers for their insightful feedback and helpful suggestions for improvement. AJ was supported by the Google PhD fellowship. JWP was supported by grants from the Simons Collaboration on the Global Brain (SCGB AWD543027), the NIH BRAIN initiative (9R01DA056404-04), an NIH R01 (1R01EY033064), and a U19 NIH-NINDS BRAIN Initiative Award (U19NS104648).

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

# A   Appendix

## A1   Relationship between a noisy EI-RNN and a CTDS

In sec. 3, we showed that in the case of no noise, it is possible to map a linear RNN with E-I structure to a CTDS under certain conditions on the connectivity of the RNN. Here, we discuss the conditions under which a noisy EI-RNN maps to a CTDS.

Following sec. 3, let us assume the following linear RNN with $N$ neurons:

$$\mathbf{y}_{t+1} = J\mathbf{y}_t + \eta_t; \eta_t \sim \mathcal{N}(0, P) \tag{A1}$$

We know from sec. 3 that for this RNN to map to a CTDS, $J = UV^\top$, where $U$ and $V$ are low rank matrices such that $U$ is non-negative block diagonal and $V$ obeys Dale's law. As a result, in the noiseless case, the CTDS that captures this model will have emission $C = U$, and the dynamics $A = V^\top U$.

In the noisy case, let's look at the joint distribution of the first two consecutive observations in a linear RNN model:

$$\begin{bmatrix} \mathbf{y}_1 \\ \mathbf{y}_2 \end{bmatrix} = \mathcal{N}\left( \begin{bmatrix} J\mathbf{y}_0 \\ J^2\mathbf{y}_0 \end{bmatrix}; \begin{bmatrix} P & JP \\ PJ^\top & JPJ^\top + P \end{bmatrix} \right) \tag{A2}$$

Next, we can also write the joint distribution of observations in any LDS model:

$$\begin{bmatrix} \mathbf{y}_1 \\ \mathbf{y}_2 \end{bmatrix} = \mathcal{N}\left( \begin{bmatrix} CA\mathbf{x}_0 \\ CA^2\mathbf{x}_0 \end{bmatrix}; \begin{bmatrix} CQC^\top + R & CAQC^\top \\ CQAC^\top & CAQA^\top C^\top + CQC^\top + R \end{bmatrix} \right) \tag{A3}$$

Now, if $C = U$ and $A = V^\top U$, the means of the observations in both settings match already. However, we need to compute the noise terms $Q$ and $R$, so that the resultant CTDS maps to a noisy EI-RNN.

Following the joint distributions of observations in an RNN and an LDS, the following three expressions should hold true:

$$P = CQC^\top + R \tag{A4}$$

$$JP = CAQC^\top \tag{A5}$$

$$JPJ^\top + P = CAQA^\top C^\top + CQC^\top + R \tag{A6}$$

$$\tag{A7}$$

Let's start with the second expression: $JP = CAQC^\top$. Since $C = U$ and $A = V^\top U$, we obtain:

$$JP = UV^\top UQU^\top \tag{A8}$$

$$UV^\top P = UV^\top UQU^\top \tag{A9}$$

$$Q = U^\dagger P U^{\dagger\top} \tag{A10}$$

Since $U$ is low-rank, not all of the covariance of $P$ will be captured by $Q$. Thus, $R$ should capture the remaining noise:

$$R = (I - UU^\top)(I - UU^\top)^\top P(I - UU^\top)(I - UU^\top)^\top \tag{A11}$$

Thus, $JP = CAQC^\top$ is now satisfied.

However, we also want:

$$P = CQC^\top + R = UQU^\top + R \tag{A12}$$

Thus, for a linear RNN to be perfectly captured by a CTDS, the eigenvectors of $P$ should either be aligned to the space spanned by $U$ or to the space orthogonal to it.

Finally, the third remaining equation has stricter implications:

$$JPJ^\top = CAQA^\top C^\top \tag{A13}$$

$$(UV^\top)P(VU^\top) = UV^\top UQU^\top VU^\top \tag{A14}$$

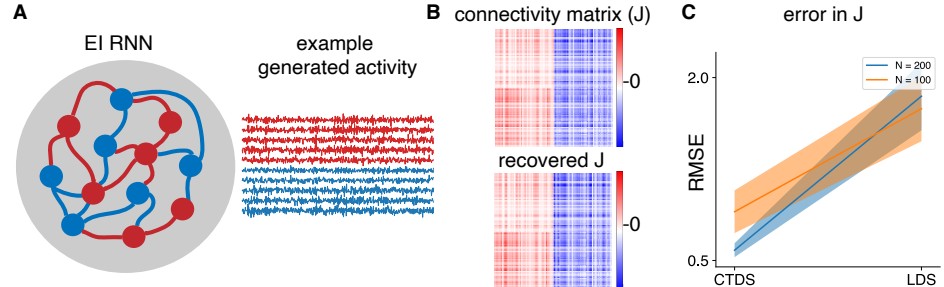

Figure A1: Simulations with an E-I network. **A** Schematic of an E-I RNN, with red units representing excitatory cells and blue units representing inhibitory cells. Example generated activity from 10 randomly selected neurons. **B**. Top shows the true connectivity matrix $J$, while the bottom plot shows the $J$ recovered using a CTDS. **C**. Root mean squared error between true and recovered $J$ using a standard LDS and a CTDS model, each of latent state dimensionality 4. We fitted them to data from two E-I networks with 100 and 200 units each. The shaded region represents 95% confidence interval.

$UQU^\top$ is low-rank and thus cannot capture $P$ entirely—it only captures covariance in the subspace spanned by $U$. Hence, for the LHS and RHS to be equal, we need $U$ and $V^\top$ to have aligned subspaces so that the effect of projecting $P$ on $V^\top$ results in the same covariance as that on the RHS.

So, in summary, we are able to find a restricted linear low-rank RNN that can be perfectly captured by a CTDS. Specifically, we want the connectivity $J$ to be expressible as $UV^\top$, with rank $D$. Here, $V$ should be in the column-space of $U$ and obey Dale's law, while $U$ should be positive block diagonal. Finally, we require that the noise covariance $P$ has eigenvectors either completely aligned with the $U$ subspace or entirely orthogonal to this subspace. In such a case, we have an CTDS that perfectly captures the activity of the RNN with the following parameters:

$$C = U, \ A = V^\top U \tag{A15}$$

$$Q = U^\dagger P U^{\dagger\top} \tag{A16}$$

$$R = (I - UU^\top)(I - UU^\top)^\top P(I - UU^\top)(I - UU^\top)^\top \tag{A17}$$

## A2 Simulations with an E-I RNN

In order to illustrate the mapping between an EI-RNN and a CTDS, we generated simulated activity from two E-I networks with 100 and 200 units respectively. The connectivity matrix, $J$, had a non-negative rank of 2 for each of the sub-matrices formed using the excitatory and inhibitory rows. In each case, the connectivity matrix $J$ was constrained to obey Dale's law, and was normalized to have eigenvalues less than 1. We set the noise matrix $P$ in accordance with sec. A1.

We then generated 10 trials of 1000 time steps each from each network (Fig. A1 A), and fitted both standard LDS and CTDS models of latent-space dimensionality 4 to the generated activity.

We initialized both models randomly, and computed recovered connectivity matrices post-fitting as follows:

$$J = CA\Sigma_\infty C^\top \left(C\Sigma_\infty C^\top + R\right)^{-1} \tag{A18}$$

We solve the lyapunov equation $\Sigma_t = A\Sigma_{t-1}A^\top + Q$ to obtain $\Sigma_\infty$. As shown in Fig. A1C, we found that the CTDS model recovers the connectivity matrix in both settings accurately, while a standard LDS does much worse in terms of root mean squared error. Fig. A1B shows the true and recovered connectivity matrices.

## A3 Assigning cell-types to neurons in FOF and ADS

In this section, we discuss how we assigned cell-types to neurons in ADS and FOF. The ADS is a part of the striatum which is known to have primarily inhibitory neurons [9], hence we assumed

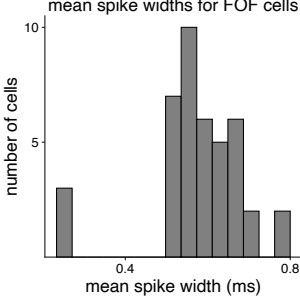

Figure A2: Histogram of the spike widths of FOF neurons. We see a bimodal distribution.

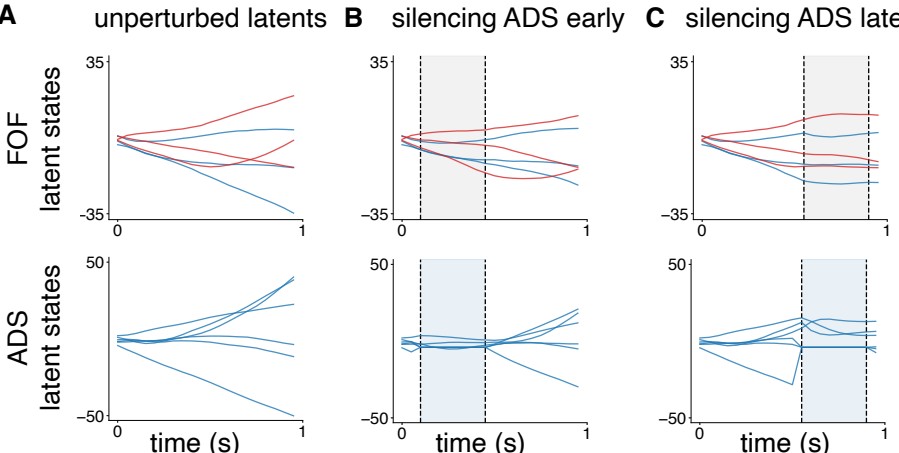

Figure A3: Latent states of fitted LDS during ADS perturbations. **A.** *(Top)* Trial-averaged unperturbed FOF latents, with randomly selected E (red) and I (blue) latents. *(Bottom)* Trial-averaged unperturbed ADS latents. **B.** *(Top)* FOF latents and *(bottom)* ADS latents, both trial-averaged, when ADS latents are silenced during the early half of trials. **C.** *(Top)* FOF latents and *(bottom)* ADS latents, both trial-averaged, with ADS silenced during the later half of trials.

all ADS neurons to be I cells. To identify cell types in the FOF, we constructed a histogram of the average spike widths of neurons in FOF. We found this to be a bimodal distribution (Fig. A2, and thus we labeled the neurons that had spike width less than $0.4$ms as inhibitory and the remaining as excitatory. This is also consistent with the known distribution of neuronal cell types in this region (20% inhibitory, 80% excitatory).

## A4 Inactivation strength during in-silico perturbation experiments

As discussed in sec. 4.3, we performed in-silico optogenetic perturbation experiments in a fitted CTDS model by clamping latents corresponding to E cells in FOF, and those corresponding to I cells in ADS. In order to choose the inactivation strength for clamping the latent states, we varied the magnitude of the latents between $[-1, 10]$ during the perturbation time steps (either the first half of a trial, or the later half of trial). Since the model was fit on neural data that was processed to have a mean firing rate of $0$Hz for every neuron, to ensure that clamping of latents indeed resulted in the inactivation of the corresponding neurons, we wanted their median firing rate to be negative. Hence, we chose the inactivation magnitude that resulted in a median firing rate of at least $-1$Hz across all neurons being perturbed. This resulted in an inactivation strength of $-2$ for FOF E latents, and $-4$ for ADS I latents (Fig. 3 shows results for these values). We did not tune the inactivation strength to match the ipsilateral bias from the experimental works [14, 37], as different studies used distinct inactivation techniques.

## A5 LDS Latent space visualization during in-silico perturbations

Finally, we also visualized the latent trajectories of a standard multi-region LDS model when performing in-silico perturbations (Fig. A3). Here, we inactivated ADS latents during either early half or late half of trials. While in a multi-region CTDS model, inactivations in ADS derailed the latent dynamics entirely (Fig. 5), we found that in a standard LDS the latent trajectories recovered quickly after perturbations during both halves of trials (Fig. A3). This is likely because CTDS constrained the connectivity between neurons and regions based on structural constraints in the brain, while the LDS did not have any such constraints. As a result, CTDS is able to accurately capture the dynamical effects of perturbation experiments, while LDS is unable to provide an accurate representation of underlying brain dynamics.

## A6 Compute requirements and code

Finally, we trained these models on 2.8 GHz Intel Cascade Lake with 8 CPU cores, and developed our code on top of [19].

