# OpenReview forum: "Disentangling the Roles of Distinct Cell Classes with Cell-Type Dynamical Systems"
_NeurIPS.cc/2024/Conference — NeurIPS 2024 spotlight_

### Official Review · Reviewer_upoR · 2024-07-11

**Soundness:** 3
**Presentation:** 3
**Contribution:** 3
**Rating:** 6
**Confidence:** 2

**Summary:**

The author developed a model named Cell-Type Dynamical Systems (CTDS) to capture the excitatory (E) and inhibitory (I) neurons’ electrical activity of rat frontal orienting fields (FOF) and anterior dorsal striatum (ADS) during an auditory decision-making task.

**Strengths:**

The experiments are well-designed and comprehensive. The results are well presented. The author systematically explores the CTDS models’ performance on E and I neurons’ electrical activity as well as these two neurons’ activity within two brain regions. The model shows higher accuracy in predicting neural activity compared to standard LDS models. By separating latent for E and I neurons, the model provides deeper insights into the functional roles of different cell classes.

**Weaknesses:**

1.	It is hard to interpret the Figure 2D; what is the y-axis of the plot for eigenvalues?
2.	For Figure 2E, why is the ADS, E latents not plotted?
3.	What is the dashed line in the ipsilateral bias plot in Figure 3CD? What does the star mean in the plot, especially in Figure 3D?

**Questions:**

The author focuses on two neuron cell types: excitatory (E) and inhibitory (I) neurons. Is it possible to extend it to other neuron cell types?
How does the CTDS model perform on neural recordings from other brain regions and tasks?
Is the CTDS model feasible for real-time applications, such as closed-loop experiments where neural activity needs to be decoded and perturbed in real time?
From figure 2E, the author stated that ‘both E and I latents in the FOF encoded the animal’s choice, with their trajectories separating out for left and right choices in opposite directions’. Does this mean only I latent of the ADS region participating in encoding the animal’s choice? But from Figure 2D, these two regions are recurrently connected. Then, what is the function of the E neurons in ADS during the decision-making process?

**Limitations:**

Addressed limitation.

---

> ### Author Rebuttal · Authors · 2024-08-06
>
> We thank the reviewer for their positive assessment of our work. We are glad that the reviewer found our work to be well presented, and our experiments to be comprehensive. Below are out detailed responses to the reviewers questions:
>
> 1. **Clarifications on figures 2D / 3CD:** Thanks for bringing these to our attention, we will add detailed captions to clarify these figures. In Fig 2D, the y-axis represents the magnitude of the eigenvalues. In Fig 3CD, the dashed line shows 0% bias, which means that the animal makes perfect choices. In Fig 3D, the stars indicate that the experimentally observed ipsilateral biases during both early-half and later-half perturbations in the ADS are significantly above 0% (Yartsev et al. 2018). In Fig 3C, the observed bias is only significant during later-half FOF perturbations as indicated by the star, and not during the early-half perturbations. Furthermore, the difference between the two perturbation effects in FOF is significant (Hanks et al. 2015).
>
>
> 2. **E latents in ADS:** Thanks for pointing this out. ADS does not have E neurons, thus we thought it was not important to visualize E latents in ADS. We discuss this in sec A3 of the appendix, but will make sure to mention this in the main text to avoid confusion.
>
>
> 3. **Extension to other cell types:** Thanks also for this comment; we feel this is an important point about the generality of our model, and we have elaborated on this in more detail in our response to all reviewers.
>
>
> 4. **Performance on other brain regions and tasks:** We focused on one task in this work, and fitted our models to one dataset containing FOF and ADS neurons. However, we would like to point out that our results (Fig 3CD) are consistent with two other papers (Hanks et al. 2015 and Yartsev et al. 2018), which study the same task but using distinct datasets. Our focus here is to introduce the model, and comprehensively demonstrate its merits using the poisson clicks task. We hope future work will apply CTDS to other tasks and brain regions.
>
>
> 5. **Real-time decoding and perturbations:** CTDS is indeed capable of real-time decoding. Once the model has been fitted, during inference Kalman filtering can be used to infer underlying latent states and behavior. Thus, it is also possible to extend this framework to control the inputs being provided to the model in a closed loop while running real-time decoding. Thanks for this important comment, we will add this to the discussion section.

---

> > ### Comment · Reviewer_upoR · 2024-08-11
> >
> > I believe the authors have addressed all my concerns. I will keep the scores unchanged.

---

> > > ### Author Response · Authors · 2024-08-12
> > >
> > > Thank you for taking the time to help improve our work.

---

### Official Review · Reviewer_KBww · 2024-07-13

**Soundness:** 3
**Presentation:** 3
**Contribution:** 3
**Rating:** 6
**Confidence:** 4

**Summary:**

This work proposed the dynamical model with cell-type specific latent, especially focused on the excitatory (E) and inhibitory (I) neurons. They developed a cell-type dynamical system (CTDS), where E/I neurons will only have positive or negative effects. They apply this model into decision making takes, and CTDS outperforms standard LDS model in decoding animal choice. They also performed an in silico experiment with optogenetics stimulation to test the causal effects on behaviors. In the end, they also demonstrate CTDS could help with identifying cell types.

**Strengths:**

**Motivation**

1. Neurons with different cell types has very distinguishable roles in neural computation, while it is often ignored in dynamical modeling for neural activities. This paper focused on important problem, and model cell types with their postive/negative effects, and demonstrating the effectiveness of the proposed model.

**Results**

1. They evaluated their proposed models on extensive applications, including decoding choices, studying causal effects from in silico experiments, and inferring cell types.
2. The results on studying causal effects are very impressive, given the model is only trained on unperturbed data, that demonstrated the learned model is very generalizable to new scenarios.

**Weaknesses:**

**Evaluation**

The proposed model only compared with over-simplified linear RNNs model, adding nonlinearity to model and add more powerful baselines will add more strengths into the evaluations.

**Results**

The CTDS does not outperform simple LDS in several settings, i.e. Fig 2c, Fig 3c.

**Applications**

Knowing cell types information in advance could be difficult in many experiment setups, using anatomical information or spike width histograms could add errors into cell type information. Evaluating the robustness of the results given incorrect cell type information.

**Related works**

Relevant references on modeling cell-type specific neural dynamics and inferring cell types is missing.

[1] Learning Time-Invariant Representations for Individual Neurons from Population Dynamics. NeurIPS 2023.

[2] Transcriptomic cell type structures in vivo neuronal activity across multiple timescales. Cell Report 2023.

**Questions:**

1. What might be the potential challenges to extend this framework into nonlinear model?
2. How to extend this into more fine-grained cell type levels? subclass (vip, sst, pvalb, etc.)

**Limitations:**

No potential negative societal impact.

---

> ### Author Rebuttal · Authors · 2024-08-06
>
> We thank the reviewer for their positive assessment of our paper and constructive suggestions for improving it. We are glad that the reviewer found our experiments to be extensive, and our causal perturbation results to be impressive. However, we feel that the reviewer has misunderstood one of our key findings, as we explain below. We apologize for not making this point clearer, and would humbly request that the reviewer consider raising their score if indeed they are persuaded by our response (on that and several other issues that we describe below).
>
> 1. **Comparison between CTDS and a standard LDS (Fig 2C, 3C):** We would like to clarify the takeaway messages from both these figures, and want to emphasize that they do not suggest that LDS models outperform CTDS. Fig 2C shows a comparison between choice accuracy as decoded from CTDS and LDS models. Indeed, both models perform equally well, however CTDS provides greater interpretability by allowing us to characterize the information encoded by the different  cell types. Furthermore, CTDS has higher test-LL which means it in fact captures neural activity __better__ than LDS.
> More importantly, in Fig 3C (left panel), we show the classification error when perturbing FOF during either half of a trial. Experimental perturbation results from rats show that this error is low during early-half perturbations, but high during late-half perturbations (see Fig 3C right panel). CTDS is indeed able to capture this phenomenon due to the distinction between E and I latents, while a regular LDS does __not__ recapitulate these findings. The fact that the “choice error” bar for the LDS model is lower than that of the CTDS model in Fig 3C does not mean that the LDS model is performing better — in fact, the opposite is true, since the point of this figure is to show that CTDS model captures the qualitative pattern of errors found in real behavior experiments (from Hanks et al. 2015, shown in the right panel of Fig 3C) better than LDS. We apologize for the confusion on this point, and we will rewrite the figure caption and text to make this clear. This is a key contribution of our work, so we hope this alleviates the reviewer’s concern.
>
>
> 2. **Comparisons to nonlinear RNNs, incorporating non-linearity:** Our rationale for showing an equivalence between CTDS and linear RNNs was to illustrate that CTDS acts as a bridge between mechanistic and descriptive models. That is, low-rank linear E-I RNNs can be exactly transformed to a CTDS model.  We agree with the reviewer, however, that non-linear RNNs are more expressive. In future work we would certainly like to extend CTDS to incorporate nonlinearities, which we agree would make it both more expressive and more biologically plausible; we will certainly add this point to the Discussion.  However, we feel that the idea of incorporating cell types into LDS models, along with the empirical results on perturbation experiments (and the  mathematical connection to linear RNNs) nevertheless make for a worthwhile contribution in their own right, which we hope the reviewer will agree on.
>
>
> 3. **Evaluating the robustness of the model using incorrect cell-type information:** This is a great question, and we thank the reviewer for bringing this up. As the reviewer points out, it may be difficult to brain accurate cell-type information. Sec 5 of our paper which focuses on inferring cell type identities gets at this question: we mask the identities of up to 50% neurons in our dataset, and fit CTDS to these datasets while also inferring neuron identity. As shown in Fig 4, CTDS is able to infer neuron identity significantly above chance for up to 50% of neurons. To answer the reviewer’s question more directly, we have also attached a figure showing test log-likelihood of CTDS fitted to datasets with varying numbers of masked neurons, as compared to when we know all cell identities accurately (in the pdf attached with author rebuttal). As we can see, while test LL expectedly falls off as the percentage of masked neurons increases, it is robust when masking up to 20% of neurons. We will add this new result to our revised manuscript.
>
> 4. **Relevant literature on inferring cell classes:** Thank you for pointing us to these papers, we will be sure to cite and discuss them in the updated version of the manuscript.
>
> 5. **Adding more fine-grained cell information:** This is an important suggestion. Please see response to all reviewers for our comments on this.

---

> > ### Comment · Reviewer_KBww · 2024-08-13
> >
> > Thanks the authors for their clarifications between comparisons between LDS and CTDS, that address my concerns, I have raised the score correspondingly.

---

> > > ### Author Response · Authors · 2024-08-13
> > >
> > > Thank you for acknowledging our response.

---

### Official Review · Reviewer_yzj8 · 2024-07-13

**Soundness:** 4
**Presentation:** 4
**Contribution:** 3
**Rating:** 7
**Confidence:** 4

**Summary:**

This work describes how a latent dynamical systems (LDS) model of neural activity can incorporate distinct excitatory (E) and inhibitory (I) latents. Doing so requires being careful to maintain sign constraints (Dale's law) in the transition matrix defining latent dynamics as well as the emission matrix defining how recorded neural activity is a projection of the underlying latents. This model can also be extended to a multi-area variant, enabling separate interaction between and within areas. The model, named a "cell-type dynamical system" (CTDS), can be fit to neural activity using an expectation-maximization inference procedure.

The model is extensively evaluated, and shown to be better than vanilla LDS. On neuropixel data from a rodent auditory decision-making task, the model predicts neural activity and behavioral choice better. Simulated optogenetic perturbations on the model better predicts behavioral effects of real perturbations. Finally, the model can be used to infer putative E/I identities of recorded neurons.

**Strengths:**

**Significance**: Latent dynamical systems are a widespread model used to explain neural activity, and this work takes the necessary step of incorporating E/I information to make these models more biologically realistic and closer to a mechanistic model. As such, this work tackles an important problem for the neuroscience community. The model is also simple and elegant, which should help its adoption.

**Novelty**: The model seems novel enough, though sign constraints is a common feature of computational neuroscience models, so it's surprising if such an approach hasn't been explored. (Can the authors better contextualize this in the Introduction?) Nevertheless, even if that were the case, such convergence is appreciated.

**Technical Quality**: The work is technically solid, with an impressively thorough evaluation on simulated data / RNN theory, predictions for neural/behavioral data, perturbations for behavior effects, and inference of cell types.

**Presentation Quality**: The writing, the mathematical notation, and the figures are all very well done.

**Weaknesses:**

1. **Cell types**: The name "cell-type dynamical systems" seems to be overselling a bit. It could more accurately be called "sign-constrained latent dynamical systems" or "latent dynamical systems with Dale's Law". Of course, the notion of "cell types" (e.g. Pyramidal, SST, PV, VIP, Purkinje) includes the excitatory/inhibitory distinction, but this is just one of many functional properties (morphology, intrinsic dynamics, stereotyped connectivity, plasticity rules, etc.) that are widely accepted as important to the distinction of cell types. Crucially, while it's relatively straightforward to incorporate sign constraints into the LDS framework as demonstrated in this work, it is not obvious to me how to incorporate these other properties that characterize cell types. If the authors feel that CTDS can indeed easily capture such properties, I would be interested to see a discussion here and in the paper about how such extensions can be done.

2. **Limitations**: I didn't see limitations in the Discussion section as mentioned in the Checklist. Please include a separate subsection to discuss this explicitly. In particular, I'd be interested to understand: (a) How the dimensionalities of the latent population are chosen for different tasks, (b) How this framework can scale to more difficult and naturalistic tasks than 2AFC.

3. **Presentation tweaks**: A couple minor revisions to improve the manuscript. Equation 1 and 3 should use a similar notation/definition for the error term. Figures use should use the same hues/shades of red and blue to denote excitatory and inhibitory data.

**Questions:**

See above.

**Limitations:**

I didn't see limitations in the Discussion section as mentioned in the Checklist.

---

> ### Author Rebuttal · Authors · 2024-08-06
>
> We thank the reviewer for their detailed  and insightful review. We are grateful that the reviewer found our work to be novel, technically solid, and of significance to the neuroscience community. Below are our responses to the reviewer’s comments and questions:
>
> 1. **Relevant literature on sign constraints in neuroscience models**:  We agree with the reviewer that a large number of previous models have sought to incorporate Dale’s law and or sign-constrained weights.  However, we would like to point out that past works have focused on incorporating Dale’s law in recurrent neural network models (RNNs), but NOT (as far as we are aware) into latent dynamical system models. For example, [Fisher et al. 2013, Haber & Schneidman 2022] have used sign-constrained RNNs to study properties of neural circuits, and [O’Shea & Duncker et al., 2022] have used them to study perturbation experiments . We have cited these works in Sec 3 of our paper, but will also mention them in the introduction for better contextualization per the reviewer’s suggestion.
> However, we are not aware of similar literature in the context of latent dynamical systems. Latent dynamical systems have typically focused on descriptive representations of neural activity, while sidestepping mechanistic properties of the brain. Our work, thus, bridges this gap by enforcing mechanistic constraints on linear dynamical systems. We thank the reviewer for bringing this up, and we will make this clear in the introduction of our revised manuscript.
>
>
> 2. **Extension to other cell types, model nomenclature:** We agree with the reviewer that here we only focus on E and I cell types, and have not discussed more fine-grain cell identities.  However, we feel that our framework can easily incorporate multiple cell types within a single population or across multiple populations (as discussed in the example case of PV and SOM inhibitory neurons in the response to all reviewers). We apologize for not making this generality clear in our original paper, and will revise both the Introduction and Discussion sections to discuss specific ways in which CTDS can be applied to multiple cell types (and not just E and I).
>
>
> 3. **Limitations section:** Thanks for pointing this out. We will add an explicit limitations section in the manuscript discussing choice of latent dimension, and scaling to other tasks. Currently, we choose latent space dimensionality based on test log-likelihood as well as for ease of visualization / interpretation. However, our results qualitatively hold for a wide range of latent dimensions (we will add this in the appendix).
> In terms of applying the model to more naturalistic tasks beyond 2AFC, two main challenges would come up:
>  (a) The stimulus could be high-dimensional: this would mean learning a correspondingly higher-dimensional B matrix in our current CTDS setup. Alternatively, we could also learn a lower-dimensional encoding of the stimulus, depending on the specifics of the task.
>  (b) For naturalistic tasks spanning over larger timescales, a linear model might not be sufficient. In such cases, our approach can be extended to incorporate sign constraints in a non-linear setup such as a switching linear dynamical system.
>
>
> 4. **Presentation tweaks**: Thank you pointing these out, we have fixed these now.

---

> ### Comment · Reviewer_yzj8 · 2024-08-12
>
> Thanks for your comments and manuscript improvements.
>
> I still feel that the nomenclature of "cell-types" is overselling a bit. The authors' global response points to PV and SOM as distinct types with specific connectivity/signs that future work could model. While connectivity and sign are indeed part of distinguishing these cell types, other properties are also important (e.g. dynamics of fast-spiking in PV vs. low-threshold spiking in SOM). As noted in my review, my concern was not that distinct connectivity/signs couldn't be represented in this framework (they can, as demonstrated in the model and multi-area variant). My concern was that other important functional properties that are widely accepted as important to the distinction of cell types (especially morphology and intrinsic dynamics) can't be trivially incorporated. So, the contribution here really is "sign-constrained latents". If renaming isn't desirable at this stage, then perhaps a discussion in the paper can highlight that this research direction is still wide open.
>
> Overall, I am excited that this work explores how to make LDS models more mechanistic by incorporating biological properties, a contribution the authors have reiterated in their response. I am happy to maintain my score to accept.

---

> > ### Author Response · Authors · 2024-08-12
> >
> > Thank you for your response, and for your insightful suggestions. We will certainly add a discussion about the limitations of our current approach in terms of capturing properties of different cell types. We agree with the reviewer that there is room for future work in this space, at the same time we also think that our model provides a valuable framework to disentangle roles of different cell types. Thanks again for your encouragement!

---

### Author Rebuttal · Authors · 2024-08-06

We thank the reviewers for their positive assessment of our work, and for their detailed comments and suggestions. We are delighted that the reviewers found our work to be well-written, of significance to the community, and our experiments to be well-designed and technically solid. We thank reviewers yzj8 and upoR for placing our work above the acceptance threshold. We also thank reviewer KBww for their constructive feedback; we have done our best to address their concerns in our detailed rebuttal below, and would humbly request that they consider raising their score.

We will discuss a general point mentioned by all reviewers first, and then address reviewer-specific comments.

**Incorporating additional cell types beyond E and I**: As the reviewers have pointed out, our paper focuses primarily on populations with two cell classes: excitatory and inhibitory neurons. Indeed, neural populations can be divided into more fine-grain cell-types, and we intend for our model to be applicable to these cell-types. For example, for a population with two distinct classes of inhibitory neurons (e.g., PV and SOM), we could use the CTDS modeling framework to incorporate distinct latent variables for each of these two populations, and the dynamics matrix could be structured to assume for particular forms of connectivity between these populations as determined by anatomy (e.g., that both PV and SOM cell types inhibit excitatory neurons, and that SOM neurons inhibit PV, but not vice versa).  However, we realize that we did not make this point clear in our original submission, and we will clarify the generality of our framework in the Introduction and Discussion sections of the revised manuscript.

Note that we have already considered a case with two different regions, where the projection from one region to the other is purely excitatory, whereas the other allows connections of both signs (once again, as suggested by anatomy).  We therefore prefer to keep the name “Cell Type Dynamical Systems” for our framework, although we are open to additional suggestions if the reviewers feel strongly.  We hope that future work can use our approach to disentangle information encoded by a range of different cell types.

---

### Decision · Program_Chairs · 2024-09-25

**Decision:**

Accept (spotlight)

**Comment:**

This paper proposes a novel latent dynamical system model for neural data that can distinguish between cell types. The reviewers unanimously agree that the paper is innovative and technically solid and therefore suitable for NeurIPS. However, two reviewers raised concerns that, although the title and abstract broadly refer to cell types, the proposed method mainly concerns excitatory and inhibitory cells. It remains unclear whether the method would generalize to other cell types. Aside from this, based on my reading of the discussions, most of the reviewers' comments have either been addressed or are not seen as barriers to the paper's acceptance at NeurIPS.